# OADAT: Experimental and Synthetic Clinical Optoacoustic Data for Standardized Image Processing

**Firat Ozdemir**[1*]     **Berkan Lafci**[2,3*]     **Xosé Luís Deán-Ben**[2,3]

**Daniel Razansky**[2,3]     **Fernando Perez-Cruz**[1,4]

[1]*Swiss Data Science Center, ETH Zurich and EPFL, Zurich, Switzerland*
[2]*Institute of Pharmacology and Toxicology and Institute for Biomedical Engineering, Faculty of Medicine, University of Zurich, Switzerland*
[3]*Institute for Biomedical Engineering, Department of Information Technology and Electrical Engineering, ETH Zurich, Switzerland*
[4]*Institute for Machine Learning, Department of Computer Science, ETH Zurich, Switzerland*

**Reviewed on OpenReview:** *https: // openreview. net/ forum? id=BVi6MhKOOG*

## Abstract

Optoacoustic (OA) imaging is based on excitation of biological tissues with nanosecond-duration laser pulses followed by subsequent detection of ultrasound waves generated via light-absorption-mediated thermoelastic expansion. OA imaging features a powerful combination between rich optical contrast and high resolution in deep tissues. This enabled the exploration of a number of attractive new applications both in clinical and laboratory settings. However, no standardized datasets generated with different types of experimental set-up and associated processing methods are available to facilitate advances in broader applications of OA in clinical settings. This complicates an objective comparison between new and established data processing methods, often leading to qualitative results and arbitrary interpretations of the data. In this paper, we provide both experimental and synthetic OA raw signals and reconstructed image domain datasets rendered with different experimental parameters and tomographic acquisition geometries. We further provide trained neural networks to tackle three important challenges related to OA image processing, namely accurate reconstruction under limited view tomographic conditions, removal of spatial undersampling artifacts and anatomical segmentation for improved image reconstruction. Specifically, we define 44 experiments corresponding to the aforementioned challenges as benchmarks to be used as a reference for the development of more advanced processing methods.

## 1 Introduction

Optoacoustic (OA) imaging is being established as a powerful method with increasing application areas in clinical (Steinberg et al., 2019; Su et al., 2010) and preclinical settings (Lafci et al., 2020; Merčep et al., 2015). Using nanosecond-duration pulsed lasers operating in the visible and near-infrared (NIR) optical wavelength range, biological tissues are thermoelastically excited. This excitation yields ultrasound (US) waves, from which OA images are tomographically reconstructed (Fig. 1a). The rich optical contrast from endogenous tissue chromophores such as blood, melanin, lipids and others are combined with high US resolution, i.e., tens of micrometers. This unique feature makes OA particularly suitable for molecular and functional imaging. Other important advantages such as the feasibility of hand-held operation, the fast acquisition performance (real time feedback) and the non-invasive safe contrast (i.e., non-ionizing radiation) further foster the wide

---

*Equal contribution. Author ordering determined by coin flip.
 Correspondence: firat.ozdemir@datascience.ch, berkan.lafci@uzh.ch

use of OA in multiple biomedical studies. OA imaging has been shown to provide unique capabilities in studies with disease models e.g., of breast cancer (Manohar & Dantuma, 2019; Diot et al., 2017; Butler et al., 2018), as well as for the clinical assessment of Crohn's disease (Knieling et al., 2017), atherosclerotic carotid plaques (Karlas et al., 2021) or skin cancer (Deán-Ben & Razansky, 2021). As the range of applications of OA imaging gets broader, the need for different data processing pipelines increases in parallel. Also, new methods are continuously being developed to provide an enhanced OA performance. Specific examples include increased temporal resolution with compressed/sparse data acquisitions (Özbek et al., 2018), accurate image reconstruction algorithms (Deán-Ben et al., 2012), light fluence correction by segmenting the tissue boundaries (Lafci et al., 2021) or enhanced spectral unmixing algorithms from multispectral data (Tzoumas & Ntziachristos, 2017).

Three major challenges suitable for data-driven approaches in clinical OA imaging are summarized below:
**Sparse acquisition:** OA imaging provides a unique potential to monitor fast-changing events such as cardiac arrhythmias (Çağla Özsoy et al., 2021), neuronal activity (Robin et al., 2021) or indocyanine green clearance (Grünherz et al., 2022) *in vivo*. For this, ultra-fast imaging systems capable of capturing changes in living organisms occurring at up to millisecond temporal scales are required. The main limiting factor affecting the achievable frame rate is the data transfer capacity. This limitation can be eliminated by reducing the number of acquired channels (signals). Therefore, sparse or compressed sensing methods have been proposed both using conventional methods (Özbek et al., 2018) and deep learning algorithms (Davoudi et al., 2019).
**Limited view reconstruction:** OA is inherently a tomographic imaging modality. Acquisition of pressure signals from different angles is essential to capture the information encoded in US waves traveling in a 3D medium in order to render accurate tomographic reconstructions. This further increases the image contrast, resolution and quantitativeness. However, tomographic coverage of the samples is often hindered by physical restrictions. Thereby, new image processing pipelines have been suggested to improve limited-view-associated challenges in OA imaging by using data-driven algorithms in the image domain (Guan et al., 2020), signal domain (Klimovskaia et al., 2022) and combination of both domains (Davoudi et al., 2021; Lan et al., 2019).
**Segmentation:** Optimal OA reconstruction algorithms need to account for different optical and acoustic properties in biological tissues and in the coupling medium (water). For example, the speed of sound (SoS) depends on the elastic properties of the medium. Proper assignment of SoS values in tissues and in water requires accurate delineation of the tissue boundaries. Thereby, segmentation of structures (Lafci et al., 2021; Schellenberg et al., 2022) in OA images has been shown to enhance the image reconstruction performance. Additionally, the optical fluence (intensity) also varies with depth across different tissues. This issue remains as one of the main factors affecting quantification in OA images (Brochu et al., 2017) and can also be corrected with tissue segmentation (Mandal et al., 2016).

As an emerging method, OA imaging requires standardization, open source code publication and data sharing practices to expedite the development of new application areas and data processing pipelines. Particularly, the aforementioned challenges associated to high data throughput, limited angular coverage, SoS assignment and fluence corrections require coordinated efforts between experimental and data science communities. Initial efforts to standardize data storage formats and image reconstruction algorithms have been undertaken (Gröhl et al., 2022). However, Gröhl et al. (2022) focus on standard reconstruction algorithms and propose data storage formats for acquisition related metadata. Data-driven image and signal processing methods require additional initiatives on fast access to large bulks of OA image and signal data and benchmarks for learning-based methods.

Here, we provide experimental data and simulations of forearm datasets as well as benchmark networks aiming at facilitating the development of new image processing algorithms and benchmarking. These "Experimental and Synthetic Clinical Optoacoustic Data (OADAT)" include, (i) large and varied clinical and simulated forearm datasets with paired subsampled or limited view image reconstruction counterparts, (ii) raw signal acquisition data of each such image reconstruction, (iii) definition of 44 experiments with gold standards focusing on the aforementioned OA challenges, (iv) pretrained model weights of the networks used for each task, and (v) user-friendly scripts to load and evaluate the networks on our datasets. The presented datasets and algorithms will expedite the research in OA image processing.

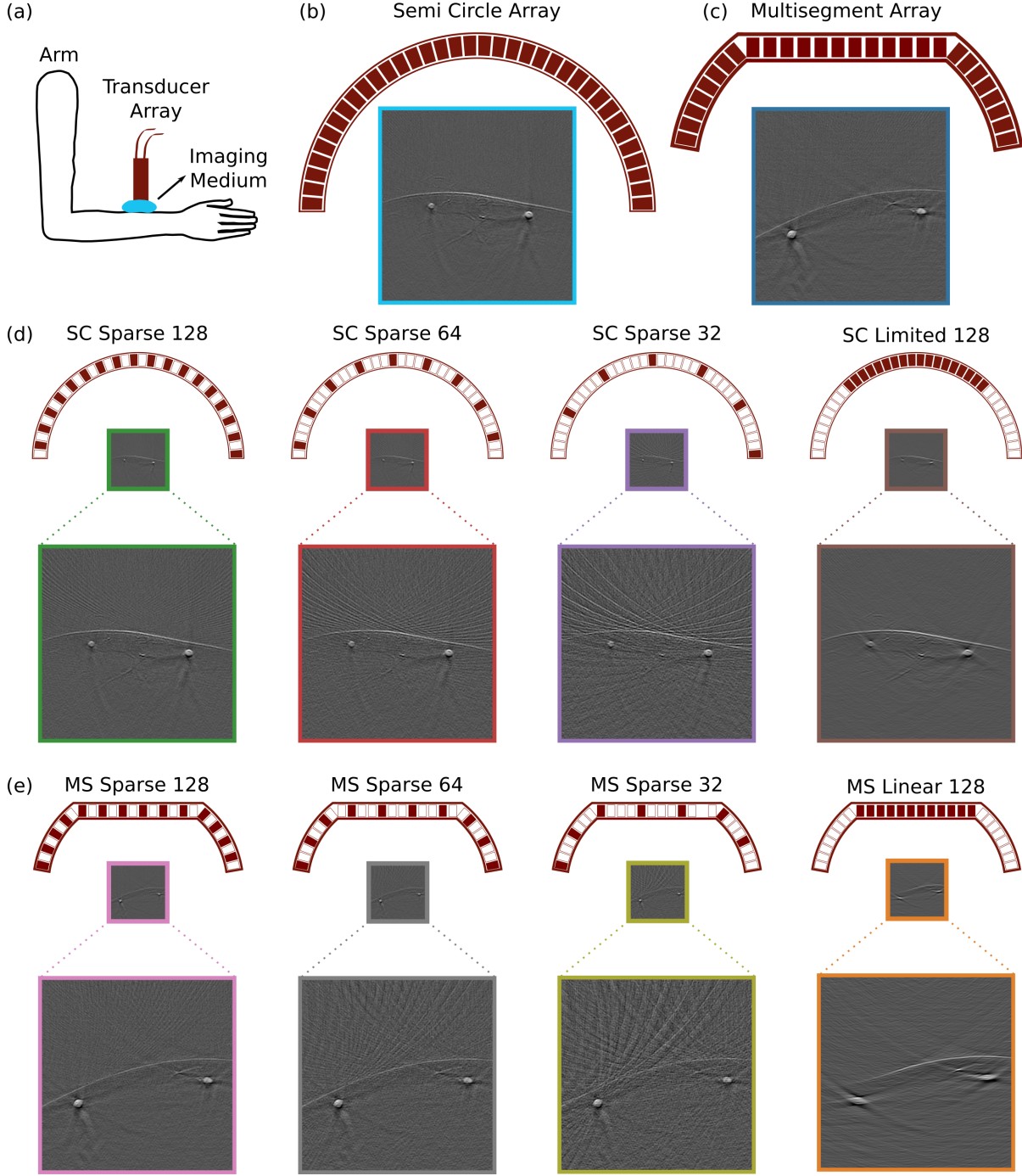

Figure 1: Experimental data acquisition, transducer arrays and resulting images. (a) Experimental setup for optoacoustic forearm imaging. (b) Semi circle array along with an example of acquired images. (c) Multisegment array along with an example of acquired images. (d) Uniform subsampling for the semi circle array (128, 64 and 32 elements) and limited view acquisition for the semi circle array with reduced angular coverage (128 elements). (e) Uniform subsampling for the multisegment array (128, 64 and 32 elements), and linear array acquisition for the multisegment array (128 elements). Transducer elements are shown as actively receiving (red) or off (white).

## 2 Background

For OA imaging, the objects are excited with the nanosecond-duration laser pulses in visible or NIR light wavelengths which result in thermoelastic expansion of the structures. This expansion generates pressure waves (US signals) that are detected by transducer arrays. Corresponding images are reconstructed by solving the OA inverse problem on the acquired signals. Below, we explain the transducer arrays used for data acquisition, the reconstruction algorithm used to generate images from acquired signals and the sampling/acquisition techniques. Detailed explanation about OA imaging and used tools can be found in Appendix A.

### 2.1 Transducer arrays

**Semi circle** array contains 256 transducers elements distributed equidistantly over a semi circle (concave surface, Fig. 1b). **Multisegment** array is a combination of linear array in the center and concave parts on the right and left sides, designed to increase angular coverage, as shown in Fig. 1c. The linear part contains 128 elements and each of the concave parts consist of 64 elements, totaling to 256. **Linear** array is the central part of the multisegment array with 128 elements (Fig. 1c). The array geometry is optimized for US data acquisitions with planar waves. Hence, it produces OA images with limited view artifacts due to reduced angular coverage which is a limiting factor for OA image acquisitions. **Virtual circle** array is generated to simulate images with 360 degree angular coverage and yields artifact free reconstructions (Fig. 2a). It contains 1,024 transducer elements distributed over a full circle with equal distance. We also have a virtual multisegment array that correspond to its physical counterpart. Additional geometric and technical details are listed in Appendix A.2.

### 2.2 Reconstruction method

We use backprojection algorithm in this study to generate OA images from the acquired signals[1]. This algorithm is based on delay and sum beamforming approach (Ozbek et al., 2013) (see Appendix A.3 for details). First, a mesh grid is created to represent the imaged field of view. Then, the distance between the points of the mesh grid and array elements are calculated based on the known locations of the transducers. Time of flight is obtained through dividing distance by the SoS values that are assigned based on the temperature of the imaging medium and tissue properties. The clinical and simulated data are reconstructed with SoS of 1,510 m/s in this study as the simulations and the experiments were done at the corresponding imaging medium temperature. Unlike more sophisticated model-based reconstruction approaches Deán-Ben et al. (2012), backprojection is parameterized using only SoS, making it a stable choice across all imaged scenes.

### 2.3 Sparse sampling

Sparse sampling yields streak artifacts on reconstructed images due to large inter-element pitch size. For a given angular coverage, i.e., transducer array geometry, using less transducer elements for reconstruction causes stronger artifacts due to increased inter-element pitch size. We define sparse sampled semi circle array acquisitions **semi circle sparse 128**, **semi circle sparse 64** and **semi circle sparse 32** when using 128, 64 and 32 elements out of the 256 of semi circle array (Fig. 1d first three columns), respectively. Similarly, we define sparse sampled virtual circle array acquisitions **virtual circle sparse 128**, **virtual circle sparse 64** and **virtual circle sparse 32** when using 128, 64 and 32 elements out of 1,024 of virtual circle array (Fig. 2c first three rows), respectively. In addition, we also define sparse sampled multisegment array acquisitions **multisegment sparse 128**, **multisegment sparse 64** and **multisegment sparse 32** when using 128, 64 and 32 elements out of the 256 elements of the multisegment array (Figs. 1e and 2d first three columns), respectively. All items correspond to uniform and hence equidistant subsampling of the corresponding transducer array signals.

---

[1]Python module for OA reconstruction: github.com/berkanlafci/pyoat.

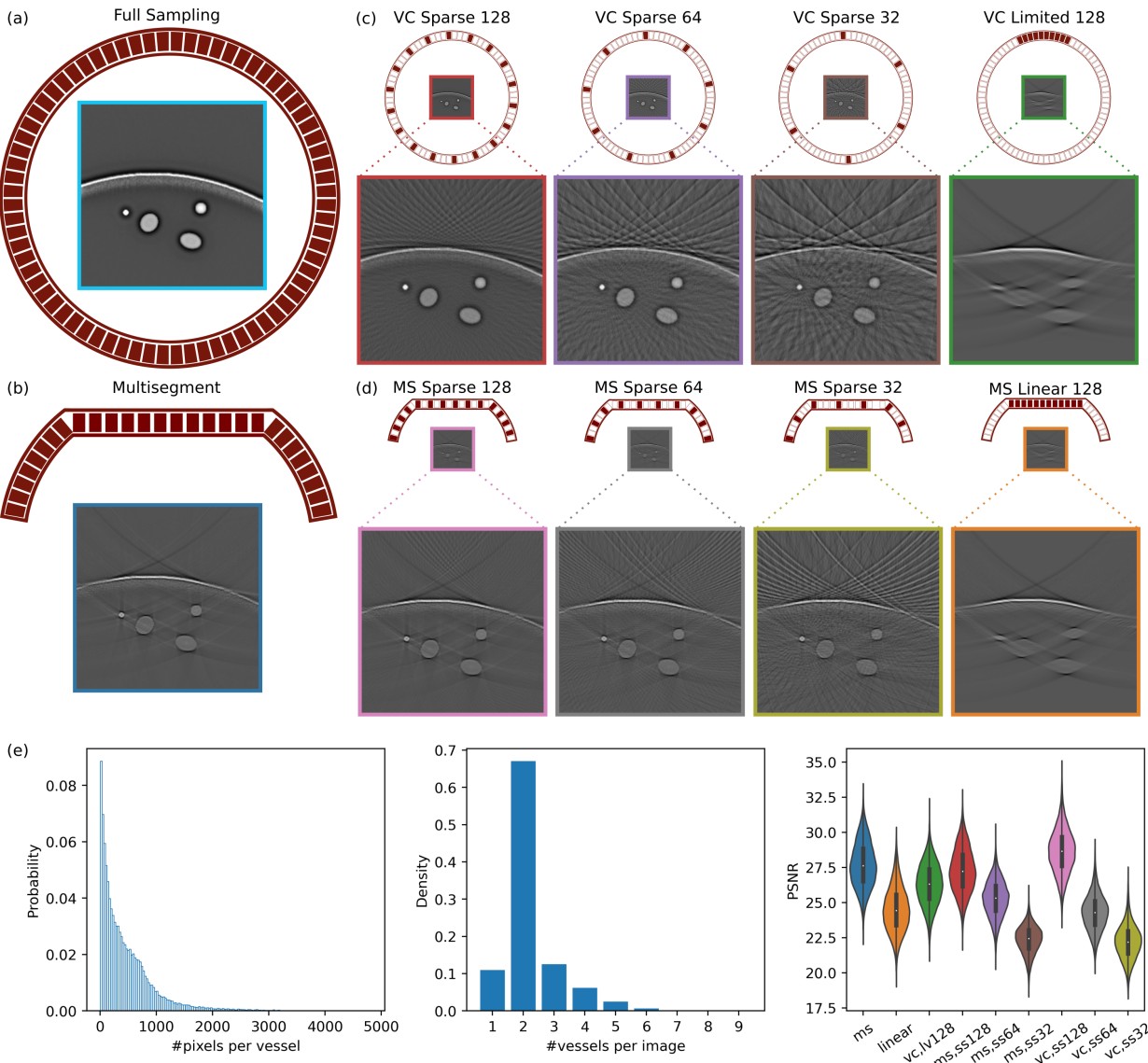

Figure 2: Overview of the simulated data. (a) Virtual circle array and an image reconstructed using 1,024 transducer elements. (b) Multisegment array and the corresponding image reconstructed using combined linear and concave parts of the transducer array. (c) Uniform subsampling of virtual circle array with 128, 64 and 32 elements and limited view acquisition with reduced angular coverage (128 elements). (d) Uniform subsampling of multisegment array with 128, 64 and 32 elements and linear array acquisition (128 elements). (e) Vessel size distribution (pixels per vessel), number of vessels per image, and peak signal-to-noise ratio of full sampling compared to other reconstructions (x axis naming conventions are explained in Sec. 3.3). Transducer elements are shown as actively receiving (red) or off (white).

Table 1: Publicly available OA datasets, supported tasks, provided data format(s), size, and content. Davoudi et al. (2019) contains 274 mice and 469 phantom slices. Huang et al. (2021) has 10 mice with 10 frames (100 slices) at 27 different wavelengths and 20 phantom slices.

| Dataset | tasks | | | image reconstruction | raw signal | size (>5k instances) | clinical data |
|---|---|---|---|---|---|---|---|
| | limited view | sparse sampling | pixel annotations | | | | |
| Davoudi et al. (2019) | ✗ | ✓ | ✗ | ✓ | ✗ | ✗ | ✗ |
| Huang et al. (2021) | ✗ | ✗ | ✗ | ✗ | ✓ | ✗ | ✗ |
| MSFD (ours) | ✓ | ✗ | ✗ | ✓ | ✓ | ✓ | ✓ |
| SWFD (ours) | ✓ | ✓ | ✗ | ✓ | ✓ | ✓ | ✓ |
| SCD (ours) | ✓ | ✓ | ✓ | ✓ | ✓ | ✓ | ✗ |
| OADAT-mini (ours) | ✓ | ✓ | ✓ | ✓ | ✓ | ✓ | ✗ ∪ ✓ |

## 2.4 Limited View

Limited view acquisitions lead to distorted geometry (e.g., elongated vessels) due to the reduced angular coverage (Figs. 1d,e & 2c,d last column, limited 128 and linear 128). To mimic commonly occurring limited view settings, we use a continuous subset of elements for a given transducer. This corresponds to retaining inter-element pitch size while reducing the angular coverage. We define a limited view acquisition for each transducer array as follows: (i) **Linear array** is the common practice in clinical settings for US data acquisition (Jensen, 2007; Luca et al., 2018). Typically, the same linear geometry is combined with OA imaging to provide complementary information (Merčep et al., 2017). To model this clinically realistic scenario, we use the linear part of the multisegment array for OA image reconstruction. (ii) **Semi circle limited 128** uses half of the semi circle array; 128 transducer elements, yielding a quarter circle. The differences between linear and semi circle limited view array acquisitions are the inter-element pitch size, focusing and geometry of the active area. (iii) **Virtual circle limited 128** uses 128 consecutive elements (one eighth of a circle) out of 1,024.

## 3 Datasets

We present four datasets [2] [3] (two experimental, one simulated, one fully annotated subset) where each has several subcategories for the purpose of tackling different challenges present in the domain. Raw signal acquisition data that is used to reconstruct all images are also provided with the datasets. Experimental datasets also include details about the volunteer Fitzpatrick skin phototype (Gupta & Sharma, 2019), which relates to the amount of melanin pigment in the skin (see Appendix B for distribution and further details). We also display a comparative overview of publicly available and our proposed OA datasets in Table 1. Please refer to Tables 5, 6, 7, 8, and 9 for summaries of the file contents of the datasets in the Appendix.

## 3.1 Multispectral forearm dataset

Multispectral forearm dataset (MSFD) is collected using multisegment array (Sec. 2.1) from nine volunteers at six different wavelengths (700, 730, 760, 780, 800, 850 nm) for both arms. Selected wavelengths are particularly aimed for spectral decomposition aiming to separate oxy- and deoxy-hemoglobin (Perekatova et al., 2016). All wavelengths are acquired consecutively, yielding almost identical scene being captured for a given slice across different wavelengths with slight displacement errors. For each of the mentioned category 1,400 slices are captured, creating a sum of $9 \times 6 \times 2 \times 1,400 = 151,200$ unique signal matrices.

From this data, using backprojection algorithm, we reconstruct (i) linear array images $\text{MSFD}_{\text{linear}}$, (ii) multisegment array images $\text{MSFD}_{\text{ms}}$, (iii) multisegment sparse 128 images (Sec.2.3), $\text{MSFD}_{\text{ms,ss128}}$, (iv) multisegment sparse 64 images (Sec.2.3), $\text{MSFD}_{\text{ms,ss64}}$, and (v) multisegment sparse 32 images (Sec.2.3),

---

[2]Link to our datasets: hdl.handle.net/20.500.11850/551512
[3]Repository for accessing and reading datasets: github.com/berkanlafci/oadat

$\mathrm{MSFD_{ms,ss32}}$, where each dataset has 151,200 images of $256 \times 256$ pixel resolution; totaling to 756,000 image instances.

## 3.2 Single wavelength forearm dataset

Single wavelength forearm dataset (SWFD) is collected using both multisegment and semi circle arrays (Sec. 2.1) from 14 volunteers at a single wavelength (1,064 nm) for both arms. The choice of the wavelength is based on maximizing penetration depth for excitation light source (laser) (Sharma et al., 2019). Out of the 14 volunteers, eight of them have also participated in the MSFD experiment and their unique identifiers match across the dataset files. For each array, volunteer, and arm, we acquired 1,400 slices, creating a sum of $2 \times 14 \times 2 \times 1,400 = 78,400$ unique signal matrices. It is important to note that despite the data being acquired from the same volunteers, signals between multisegment array and semi circle array are not paired due to physical constraints.

From this data, using backprojection algorithm, we reconstruct (i) linear array images, $\mathrm{SWFD_{linear}}$, (ii) multisegment array images, $\mathrm{SWFD_{ms}}$, (iii) semi circle array images, $\mathrm{SWFD_{sc}}$, (iv) semi circle array limited 128 images (Sec.2.4), $\mathrm{SWFD_{sc,lv128}}$, (v) semi circle sparse 128 images (Sec.2.3), $\mathrm{SWFD_{sc,ss128}}$, (vi) semi circle sparse 64 images (Sec.2.3), $\mathrm{SWFD_{sc,ss64}}$, (vii) semi circle sparse 32 images (Sec.2.3), $\mathrm{SWFD_{sc,ss32}}$, (viii) multisegment sparse 128 images (Sec.2.3), $\mathrm{SWFD_{ms,ss128}}$, (ix) multisegment sparse 64 images (Sec.2.3), $\mathrm{SWFD_{ms,ss64}}$, and (x) multisegment sparse 32 images (Sec.2.3), $\mathrm{SWFD_{ms,ss32}}$, where each dataset has 39,200 images of $256 \times 256$ pixel resolution; totaling to 392,000 image instances.

## 3.3 Simulated cylinders dataset

Simulated cylinders dataset (SCD) is a group of synthetically generated 20,000 forearm acoustic pressure maps that we heuristically produced based on a range of criteria we observed in experimental images. The acoustic pressure maps are generated with $256 \times 256$ pixel resolution where skin curves and afterwards a random amount of ellipses with different intensity profiles are generated iteratively for a given image (see Fig. 2). We explain details for the simulation algorithm[4] for generating acoustic pressure map in Appendix E.

Based on the acoustic pressure map, we generate its annotation map with three labels, corresponding to background, vessels, and skin curve. For each acoustic pressure map, we generate signal matrices for the geometries of linear, multisegment and virtual circle arrays. Using linear and multisegment array signals, we use backprojection algorithm to reconstruct (i) linear array images, $\mathrm{SCD_{linear}}$, and (ii) multisegment array images, $\mathrm{SCD_{ms}}$, (iii) multisegment sparse 128 images (Sec.2.3), $\mathrm{SCD_{ms,ss128}}$, (iv) multisegment sparse 64 images (Sec.2.3), $\mathrm{SCD_{ms,ss64}}$, and (v) multisegment sparse 32 images (Sec.2.3), $\mathrm{SCD_{ms,ss32}}$. From virtual circle array signals, we use backprojection algorithm to reconstruct (vi) virtual circle images, $\mathrm{SCD_{vc}}$, (vii) virtual circle limited 128 images (Sec.2.4), $\mathrm{SCD_{vc,lv128}}$, (viii) virtual circle sparse 128 images (Sec.2.3), $\mathrm{SCD_{vc,ss128}}$, (ix) virtual circle sparse 64 images (Sec.2.3), $\mathrm{SCD_{vc,ss64}}$, and (x) virtual circle sparse 32 images (Sec.2.3), $\mathrm{SCD_{vc,ss32}}$, where each dataset has 20,000 images of $256 \times 256$ pixel resolution; totaling to 200,000 image instances. All ten image reconstruction variations of SCD have corresponding pairs for each of the 20k image; i.e., produced from the same acoustic pressure map.

## 3.4 OADAT-mini dataset

Using a subset of 100 signal and corresponding reconstruction instances from each of the previously mentioned datasets, we also present OADAT-mini, which is a fragment of OADAT that is significantly smaller yet comprehensive for understanding the contents of OADAT. In addition, OADAT-mini contains manual annotation maps for vessels in the reconstructed images.

## 4 Tasks

Based on the datasets presented in Sec. 3, we define a list of experiments related to image translation to overcome (i) sparse sampling and (ii) limited view artifacts, and semantic segmentation of images.

---

[4]Python module for acoustic map simulation: renkulab.io/gitlab/firat.ozdemir/oa-armsim.

Table 2: List of tasks and experiments we define on MSFD, SWFD, and SCD. Experiment names are built of (i) dataset name for translation tasks or seg for segmentation task, (ii) input data and corresponding number of active array elements; sparse sampling (ss), limited view (lv), virtual circle (vc), and (iii) input array type; semi circle (sc), virtual circle (vc), linear (li), multisegment (ms). Image data and annotation maps are represented with $x$ and $y$, while predicted image and annotations are shown as $x^*$ and $y^*$.

| Limited view correction | Semantic segmentation |
|---|---|
| $f_{\mathrm{MSFD\_lv128,li}}$: $x \sim \mathrm{MSFD_{linear}} \to x^* \sim \mathrm{MSFD_{ms}}$ | $f_{\mathrm{seg\_vc,vc}}$: $x \sim \mathrm{SCD_{vc}} \to y^* \sim \mathrm{labels}$ |
| $f_{\mathrm{SWFD\_lv128,li}}$: $x \sim \mathrm{SWFD_{linear}} \to x^* \sim \mathrm{SWFD_{ms}}$ | $f_{\mathrm{seg\_lv128,li}}$: $x \sim \mathrm{SCD_{linear}} \to y^* \sim \mathrm{labels}$ |
| $f_{\mathrm{SWFD\_lv128,sc}}$: $x \sim \mathrm{SWFD_{sc,lv128}} \to x^* \sim \mathrm{SWFD_{sc}}$ | $f_{\mathrm{seg\_lv128,vc}}$: $x \sim \mathrm{SCD_{vc,lv128}} \to y^* \sim \mathrm{labels}$ |
| $f_{\mathrm{SCD\_lv128,li}}$: $x \sim \mathrm{SCD_{linear}} \to x^* \sim \mathrm{SCD_{vc}}$ | $f_{\mathrm{seg\_ss128,vc}}$: $x \sim \mathrm{SCD_{vc,ss128}} \to y^* \sim \mathrm{labels}$ |
| $f_{\mathrm{SCD\_lv128,vc}}$: $x \sim \mathrm{SCD_{vc,lv128}} \to x^* \sim \mathrm{SCD_{vc}}$ | $f_{\mathrm{seg\_ss64,vc}}$: $x \sim \mathrm{SCD_{vc,ss64}} \to y^* \sim \mathrm{labels}$ |
| $f_{\mathrm{SCD\_lv256,ms}}$: $x \sim \mathrm{SCD_{ms}} \to x^* \sim \mathrm{SCD_{vc}}$ | $f_{\mathrm{seg\_ss32,vc}}$: $x \sim \mathrm{SCD_{vc,ss32}} \to y^* \sim \mathrm{labels}$ |
| **Sparse sampling correction** | $f_{\mathrm{seg\_ss128,ms}}$: $x \sim \mathrm{SCD_{ms,ss128}} \to y^* \sim \mathrm{labels}$ |
| $f_{\mathrm{SWFD\_ss128,sc}}$: $x \sim \mathrm{SWFD_{sc,ss128}} \to x^* \sim \mathrm{SWFD_{sc}}$ | $f_{\mathrm{seg\_ss64,ms}}$: $x \sim \mathrm{SCD_{ms,ss64}} \to y^* \sim \mathrm{labels}$ |
| $f_{\mathrm{SWFD\_ss64,sc}}$: $x \sim \mathrm{SWFD_{sc,ss64}} \to x^* \sim \mathrm{SWFD_{sc}}$ | $f_{\mathrm{seg\_ss32,ms}}$: $x \sim \mathrm{SCD_{ms,ss32}} \to y^* \sim \mathrm{labels}$ |
| $f_{\mathrm{SWFD\_ss32,sc}}$: $x \sim \mathrm{SWFD_{sc,ss32}} \to x^* \sim \mathrm{SWFD_{sc}}$ | $f_{\mathrm{seg\_MSFD\_lv128,ms}}$: $x \sim \mathrm{MSFD_{linear}} \to y^* \sim \mathrm{labels}$ |
| $f_{\mathrm{SCD\_ss128,vc}}$: $x \sim \mathrm{SCD_{vc,ss128}} \to x^* \sim \mathrm{SCD_{vc}}$ | $f_{\mathrm{seg\_MSFD\_ss128,ms}}$: $x \sim \mathrm{MSFD_{ms,ss128}} \to y^* \sim \mathrm{labels}$ |
| $f_{\mathrm{SCD\_ss64,vc}}$: $x \sim \mathrm{SCD_{vc,ss64}} \to x^* \sim \mathrm{SCD_{vc}}$ | $f_{\mathrm{seg\_MSFD\_ss64,ms}}$: $x \sim \mathrm{MSFD_{ms,ss64}} \to y^* \sim \mathrm{labels}$ |
| $f_{\mathrm{SCD\_ss32,vc}}$: $x \sim \mathrm{SCD_{vc,ss32}} \to x^* \sim \mathrm{SCD_{vc}}$ | $f_{\mathrm{seg\_MSFD\_ss32,ms}}$: $x \sim \mathrm{MSFD_{ms,ss32}} \to y^* \sim \mathrm{labels}$ |
| $f_{\mathrm{SWFD\_ss128,ms}}$: $x \sim \mathrm{SWFD_{ms,ss128}} \to x^* \sim \mathrm{SWFD_{ms}}$ | $f_{\mathrm{seg\_SWFD\_lv128,ms}}$: $x \sim \mathrm{SWFD_{linear}} \to y^* \sim \mathrm{labels}$ |
| $f_{\mathrm{SWFD\_ss64,ms}}$: $x \sim \mathrm{SWFD_{ms,ss64}} \to x^* \sim \mathrm{SWFD_{ms}}$ | $f_{\mathrm{seg\_SWFD\_lv128,sc}}$: $x \sim \mathrm{SWFD_{sc,lv128}} \to y^* \sim \mathrm{labels}$ |
| $f_{\mathrm{SWFD\_ss32,ms}}$: $x \sim \mathrm{SWFD_{ms,ss32}} \to x^* \sim \mathrm{SWFD_{ms}}$ | $f_{\mathrm{seg\_SWFD\_ms,ms}}$: $x \sim \mathrm{SWFD_{ms}} \to y^* \sim \mathrm{labels}$ |
| $f_{\mathrm{SCD\_ss128,ms}}$: $x \sim \mathrm{SCD_{ms,ss128}} \to x^* \sim \mathrm{SCD_{vc}}$ | $f_{\mathrm{seg\_SWFD\_sc}}$: $x \sim \mathrm{SWFD_{sc}} \to y^* \sim \mathrm{labels}$ |
| $f_{\mathrm{SCD\_ss64,ms}}$: $x \sim \mathrm{SCD_{ms,ss64}} \to x^* \sim \mathrm{SCD_{vc}}$ | $f_{\mathrm{seg\_SWFD\_ss128,sc}}$: $x \sim \mathrm{SWFD_{sc,ss128}} \to y^* \sim \mathrm{labels}$ |
| $f_{\mathrm{SCD\_ss32,ms}}$: $x \sim \mathrm{SCD_{ms,ss32}} \to x^* \sim \mathrm{SCD_{vc}}$ | $f_{\mathrm{seg\_SWFD\_ss64,sc}}$: $x \sim \mathrm{SWFD_{sc,ss64}} \to y^* \sim \mathrm{labels}$ |
| $f_{\mathrm{MSFD\_ss128,ms}}$: $x \sim \mathrm{MSFD_{ms,ss128}} \to x^* \sim \mathrm{MSFD_{ms}}$ | $f_{\mathrm{seg\_SWFD\_ss32,sc}}$: $x \sim \mathrm{SWFD_{sc,ss32}} \to y^* \sim \mathrm{labels}$ |
| $f_{\mathrm{MSFD\_ss64,ms}}$: $x \sim \mathrm{MSFD_{ms,ss64}} \to x^* \sim \mathrm{MSFD_{ms}}$ | $f_{\mathrm{seg\_SWFD\_ss128,ms}}$: $x \sim \mathrm{SWFD_{ms,ss128}} \to y^* \sim \mathrm{labels}$ |
| $f_{\mathrm{MSFD\_ss32,ms}}$: $x \sim \mathrm{MSFD_{ms,ss32}} \to x^* \sim \mathrm{MSFD_{ms}}$ | $f_{\mathrm{seg\_SWFD\_ss64,ms}}$: $x \sim \mathrm{SWFD_{ms,ss64}} \to y^* \sim \mathrm{labels}$ |
| | $f_{\mathrm{seg\_SWFD\_ss32,ms}}$: $x \sim \mathrm{SWFD_{ms,ss32}} \to y^* \sim \mathrm{labels}$ |

## 4.1 Image translation

Through a list of permutations of our datasets, we can define several pairs of varying difficulty of image translation experiments where the target images are also available (see Table 2). We present sparse sampling and limited view reconstructions of SWFD, MSFD and SCD for all transducer arrays. Sparse sampling correction experiments learn mapping functions listed in Table 2, where the function notations denote the dataset used, the task of sparse sampling (ss) correction from the given number of elements used for image reconstruction and the array that is used to generate the input. Limited view correction experiments learn mapping functions, listed in Table 2, where the function notations denote the dataset used, the task of limited view (lv) correction from the given number of elements used for image reconstruction and the array that is used to generate the input.

## 4.2 Semantic segmentation

SCD includes pixel annotations for skin curve, vessels and background. In addition to segmentation of these structures on the ideal reconstructions $\mathrm{SCD_{vc}}$, we define this task on sparse sampling and limited view reconstructions that contain the relevant artifacts encountered in experimental data. Accordingly, we compose the nine segmentation experiments listed in Table 2, where the function notations denote the task segmentation (seg), type of the reconstructed input being used (virtual circle (vc), limited view (lv), and sparse sampling (ss)) and the array that is used to generate the input. All data is generated from SCD and the objective is to match the ground truth annotations of the acoustic pressure map. Different than SCD, experimental datasets under OADAT-mini include pixel annotations for vessels and consist of 14 segmentation experiments, also listed in Table 2.

# 5 Experiments and results

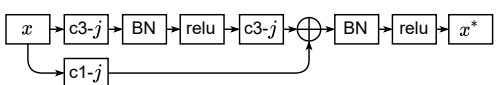

For all experiments we standardize the architecture that we based on UNet (Ronneberger et al., 2015). Specifically, we adopt the five spatial feature abstraction levels and use skip connections to concatenate with features of matching spatial dimension along the upsampling path. However, we make

Figure 3: Residual convolutional block with batch normalization (BN). c$i$-$j$ conv. layer have $i \times i$ kernels and $j$ filters.

several distinct design choices that vary from vanilla UNet. First, we use attention gates (Oktay et al., 2018) at the end of each skip connection. Second, we opt for residual convolutional blocks with batch normalization (Ioffe & Szegedy, 2015) at each level, shown in Fig. 3. Third, we use 2D bilinear upsampling instead of deconvolutions. Finally, we use half the number of convolutional kernels at each layer; e.g., start with 32 convolutional filters as opposed to 64. Full schematic as well as other implementation details are discussed in Appendix C. We refer to our modified UNet architecture as modUNet hereon.

## 5.1 Data split and preprocessing

We standardize how we split each dataset into training and test sets regardless of the task in order to ensure consistency in our and future experiments. Out of the nine volunteers in MSFD, we use five for training (IDs: 2, 5, 6, 7, 9), one for validation (ID: 10) and three for testing (IDs: 11, 14, 15). Out of the 14 volunteers in SWFD, we use eight for training (IDs: 1, 2, 3, 4, 5, 6, 7, 8), one for validation (ID: 9) and five for testing (IDs: 10, 11, 12, 13, 14). Out of the 20k slices in SCD, we use the first 14k for training, following 1k for validation, and the last 5k for testing. For each experiment conducted on OADAT-mini, we use the first 75 samples for training, next 5 for validation and the last 20 for quantitative evaluation. This translates to six times the numbers for MSFD-mini, where there are 100 samples for each of the six wavelengths. As before, we conduct the MSFD-mini segmentation experiments as a single modUNet attempting segment any of the given six wavelength samples. As a preprocessing step, all data instances (except for annotation maps) are independently scaled by their maximum and then clipped at a minimum value of $-0.2$ (Ding et al., 2015).

## 5.2 Results

We evaluate modUNet performance on the test sets using standard metrics. Namely, we report mean absolute error (MAE), root mean squared error (RMSE), structural similarity index (SSIM), and peak signal-to-noise ratio (PSNR) for image translation experiments between modUNet predictions and targets. Segmentation task performance is reported using Dice coefficient (F1-score), intersection over union (IoU, i.e., Jaccard index) and 95-percentile Hausdorff distance (HD95) metrics between modUNet predictions and annotation maps for vessels and skin curves. HD95 is calculated in several steps: First, the set of pixels along the contour for each predicted (set A) and annotated (set B) target structures are found. For each set point, the closest point from the other set is determined based on $l_2$ distance. Different from the standard Hausdorff distance, the 95-percentile distance value is taken as the directional distance from set A to B (and vice versa) instead of the maximum distance. Then the maximum of these two values is calculated as the symmetric HD95 for a given image. In OADAT-mini experiments, only vessel annotation maps are available. Since OADAT-mini

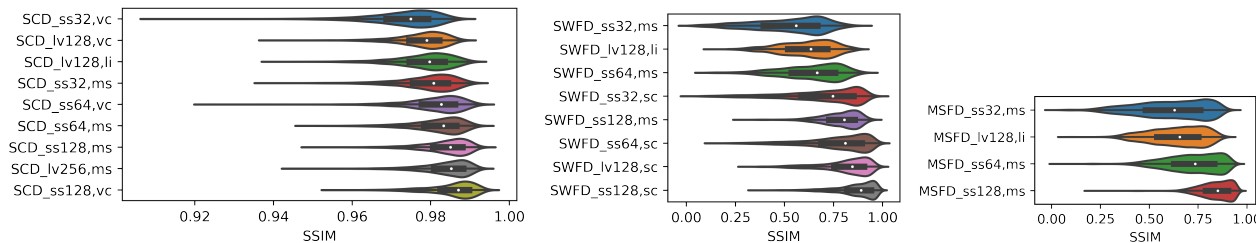

Figure 4: Distribution of modUNet SSIM performance on SCD(left), SWFD (middle) and MSFD (right) image translation experiments, sorted in ascending median test sample performance.

Table 3: Image translation results of the proposed modUNet model reported as mean ±std. Each row corresponds to the results of the experiment where input data is identified through (i) the input data and corresponding number of active transducer elements; sparse sampling (ss), limited view (lv) and (ii) the array type used for input; semi circle (sc), virtual circle (vc), linear (li), multisegment (ms).

| | | MAE | RMSE | SSIM | PSNR |
|---|---|---|---|---|---|
| **SCD** | | | | | |
| | lv128,li | 0.005 ±1.3e-3 | 0.012 ±3.3e-3 | 0.978 ±7.2e-3 | 39.909 ±2.15 |
| Limited view | lv128,vc | 0.005 ±1.3e-3 | 0.013 ±3.4e-3 | 0.978 ±6.2e-3 | 39.853 ±2.18 |
| | lv256,ms | 0.005 ±1.2e-3 | 0.009 ±3.0e-3 | 0.984 ±6.1e-3 | 42.459 ±2.52 |
| | ss128,vc | 0.004 ±1.1e-3 | 0.008 ±2.0e-3 | 0.985 ±6.1e-3 | 44.068 ±2.12 |
| | ss64,vc | 0.005 ±1.3e-3 | 0.011 ±2.9e-3 | 0.981 ±6.9e-3 | 41.402 ±2.25 |
| Sparse view | ss32,vc | 0.006 ±1.5e-3 | 0.013 ±3.7e-3 | 0.973 ±8.7e-3 | 39.309 ±2.23 |
| | ss128,ms | 0.005 ±1.1e-3 | 0.010 ±3.0e-3 | 0.984 ±6.1e-3 | 42.345 ±2.39 |
| | ss64,ms | 0.005 ±1.2e-3 | 0.010 ±3.0e-3 | 0.982 ±6.7e-3 | 41.525 ±2.19 |
| | ss32,ms | 0.005 ±1.3e-3 | 0.011 ±3.0e-3 | 0.979 ±7.3e-3 | 40.678 ±2.10 |
| **SWFD** | | | | | |
| Limited view | lv128,li | 0.028 ±1.6e-2 | 0.039 ±2.0e-2 | 0.613 ±1.4e-1 | 29.397 ±4.75 |
| | lv128,sc | 0.016 ±1.2e-2 | 0.021 ±1.5e-2 | 0.811 ±1.3e-1 | 36.791 ±5.30 |
| | ss128,sc | 0.015 ±1.2e-2 | 0.019 ±1.5e-2 | 0.863 ±1.0e-1 | 38.233 ±5.35 |
| | ss64,sc | 0.019 ±1.5e-2 | 0.024 ±1.8e-2 | 0.769 ±1.6e-1 | 35.605 ±5.22 |
| Sparse view | ss32,sc | 0.021 ±1.6e-2 | 0.028 ±2.0e-2 | 0.693 ±2.0e-1 | 33.852 ±5.37 |
| | ss128,ms | 0.023 ±1.5e-2 | 0.028 ±1.9e-2 | 0.784 ±9.7e-2 | 33.764 ±4.53 |
| | ss64,ms | 0.029 ±1.9e-2 | 0.037 ±2.3e-2 | 0.636 ±1.5e-1 | 31.311 ±4.80 |
| | ss32,ms | 0.033 ±2.1e-2 | 0.042 ±2.5e-2 | 0.521 ±1.8e-1 | 29.813 ±5.11 |
| **MSFD** | | | | | |
| limited view | lv128,li | 0.023 ±1.1e-2 | 0.035 ±1.4e-2 | 0.640 ±1.4e-1 | 29.731 ±3.97 |
| | ss128,ms | 0.017 ±9.7e-3 | 0.022 ±1.2e-2 | 0.839 ±8.1e-2 | 35.798 ±3.84 |
| Sparse view | ss64,ms | 0.022 ±1.2e-2 | 0.029 ±1.5e-2 | 0.719 ±1.4e-1 | 33.104 ±4.02 |
| | ss32,ms | 0.026 ±1.4e-2 | 0.036 ±1.7e-2 | 0.608 ±1.8e-1 | 30.873 ±4.22 |

consist of subsets of the other three datasets, we do not repeat image translation tasks. In Tables 3 & 4 we report modUNet results for mean and standard deviations aggregated over the corresponding test set images.

Using SSIM, we show performance across all our datasets in Fig. 4. Upon exploring the reason behind the long tails, we notice that most of the lower scores occur when acquisition noise and/or artifacts are more pronounced. Depending on the sample, this can imply either modUNet reduced the noise present in the target, or both input/output pair in the test set had low signal-to-noise ratio(SNR). Nevertheless, modUNet successfully corrects geometric distortions for limited view experiments. Given the low mean and standard deviations in MAE and RMSE, we can comment that modUNet can generalize well to previously unseen volunteer data. This is further corroborated with the narrow SSIM interquartile range in Fig. 4 violin plots.

Similarly, we plot the segmentation performance for IoU across different experiments on SCD in Fig. 5 for vessel and skin curve labels. While skin curve segmentation performance almost never drops below IoU of 0.80 for SCD, one can see that IoU can be drastically lower for vessel segmentation. We observed that this only happens when the size of vessel is small. For example, there are examples with ground truth vessels having as low as four pixels while the prediction has six, leading to an IoU of 0.6. As for experiments with OADAT-mini, vessel segmentation performance in experimental datasets are worse. Specifically, the worst scores are observed for MSFD-mini experiments. This is due to the movement artifact between different wavelengths of a given slice. All MSFD-mini instances have expert annotations for 800 nm wavelength. Slight movement across different wavelengths can yield poor quantitative metrics, particularly exacerbated when the observed vessels are small. Given the limited training and test sizes of OADAT-mini experiments, we believe that the quantitative results should be taken as a reference. The qualitative results in the Appendix can be

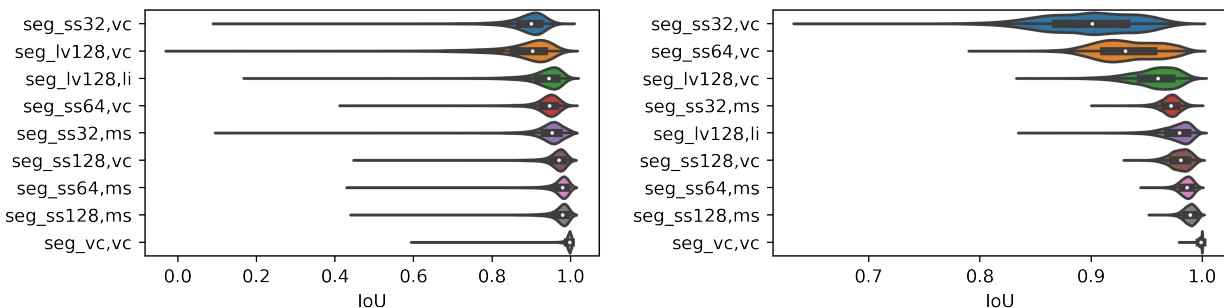

Figure 5: Distribution of modUNet IoU performance on SCD semantic segmentation experiments for vessel (left) and skin curve (right) labels, sorted in ascending median test sample performance.

Table 4: Segmentation results of our proposed modUNet model reported as mean ±std. Each row corresponds to the results of the experiment where input data is identified through (i) the input data and corresponding number of active transducer elements; sparse sampling (ss), limited view (lv), virtual circle (vc), semi circle (sc), multisegment (ms) and (ii) the array type used for input; virtual circle (vc), semi circle (sc), multisegment (ms), and linear (li). SWFD- and MSFD-mini correspond to experiment conducted on OADAT-mini dataset.

| | | Dice | | IoU | | HD95 | |
|---|---|---|---|---|---|---|---|
| | | vessels | skin curve | vessels | skin curve | vessels | skin curve |
| **SCD** | | | | | | | |
| Full view | vc,vc | 0.996 ±9.8e-3 | 0.999 ±8.7e-4 | 0.993 ±1.8e-2 | 0.999 ±1.7e-3 | 0.792 ±1.0e+0 | 0.164 ±2.0e+0 |
| Limited view | lv128,li | 0.963 ±3.4e-2 | 0.988 ±7.3e-3 | 0.931 ±5.5e-2 | 0.976 ±1.4e-2 | 2.734 ±2.9e+0 | 2.196 ±7.4e+0 |
| | lv128,vc | 0.933 ±5.8e-2 | 0.978 ±1.1e-2 | 0.878 ±8.6e-2 | 0.957 ±2.1e-2 | 4.310 ±8.9e+0 | 7.372 ±1.4e+1 |
| | ss128,vc | 0.980 ±2.3e-2 | 0.990 ±4.7e-3 | 0.961 ±4.0e-2 | 0.980 ±9.1e-3 | 2.499 ±5.2e+0 | 3.897 ±1.0e+1 |
| | ss64,vc | 0.967 ±2.8e-2 | 0.965 ±1.5e-2 | 0.937 ±4.6e-2 | 0.933 ±2.8e-2 | 2.738 ±3.0e+0 | 9.634 ±1.5e+1 |
| Sparse view | ss32,vc | 0.938 ±3.9e-2 | 0.946 ±2.3e-2 | 0.886 ±6.0e-2 | 0.899 ±4.1e-2 | 3.074 ±3.1e+0 | 12.220 ±1.6e+1 |
| | ss128,ms | 0.983 ±2.4e-2 | 0.994 ±2.9e-3 | 0.968 ±4.1e-2 | 0.989 ±5.6e-3 | 2.289 ±2.4e+0 | 5.311 ±1.3e+1 |
| | ss64,ms | 0.984 ±2.4e-2 | 0.993 ±3.1e-3 | 0.969 ±4.1e-2 | 0.985 ±6.1e-3 | 2.772 ±8.2e+0 | 2.668 ±8.5e+0 |
| | ss32,ms | 0.971 ±2.8e-2 | 0.985 ±4.8e-3 | 0.945 ±4.5e-2 | 0.971 ±9.2e-3 | 3.026 ±7.0e+0 | 6.795 ±1.5e+1 |
| **SWFD-mini** | | | | | | | |
| Full view | sc,sc | 0.857 ±5.7e-2 | N/A | 0.755 ±8.5e-2 | N/A | 17.225 ±2.8e+1 | N/A |
| | ms,ms | 0.842 ±5.9e-2 | N/A | 0.732 ±8.6e-2 | N/A | 17.218 ±2.6e+1 | N/A |
| Limited view | lv128,li | 0.794 ±6.4e-2 | N/A | 0.662 ±8.3e-2 | N/A | 28.685 ±3.9e+1 | N/A |
| | lv128,sc | 0.843 ±5.5e-2 | N/A | 0.732 ±7.8e-2 | N/A | 24.444 ±2.6e+1 | N/A |
| | ss128,sc | 0.864 ±4.3e-2 | N/A | 0.763 ±6.5e-2 | N/A | 23.146 ±3.0e+1 | N/A |
| | ss64,sc | 0.841 ±1.1e-1 | N/A | 0.737 ±1.3e-1 | N/A | 20.216 ±2.8e+1 | N/A |
| Sparse view | ss32,sc | 0.864 ±4.1e-2 | N/A | 0.762 ±6.3e-2 | N/A | 26.832 ±3.0e+1 | N/A |
| | ss128,ms | 0.836 ±5.8e-2 | N/A | 0.722 ±8.3e-2 | N/A | 17.909 ±2.8e+1 | N/A |
| | ss64,ms | 0.837 ±5.6e-2 | N/A | 0.723 ±8.0e-2 | N/A | 15.811 ±2.3e+1 | N/A |
| | ss32,ms | 0.809 ±6.0e-2 | N/A | 0.684 ±8.2e-2 | N/A | 20.603 ±3.1e+1 | N/A |
| **MSFD-mini** | | | | | | | |
| Limited view | lv128,li | 0.474 ±1.3e-1 | N/A | 0.320 ±1.2e-1 | N/A | 20.134 ±3.3e+1 | N/A |
| | ss128,ms | 0.563 ±1.0e-1 | N/A | 0.400 ±1.1e-1 | N/A | 14.312 ±1.9e+1 | N/A |
| Sparse view | ss64,ms | 0.572 ±1.2e-1 | N/A | 0.411 ±1.3e-1 | N/A | 19.716 ±2.1e+1 | N/A |
| | ss32,ms | 0.639 ±1.4e-1 | N/A | 0.485 ±1.5e-1 | N/A | 11.499 ±1.7e+1 | N/A |

more informative for gaining insight for modUNet performance when trained with a very limited amount of data. We provide qualitative results as well as conduct further analysis for all tasks in Appendix D.

## 6    Discussion

Major differences exist between simulated and experimental datasets. Even if the content is different, training and test samples of the simulated dataset are inherently sampled from the same distribution. On the other hand, experimental datasets feature shifts due to different volunteers being imaged, inherent noise from data acquisition system, and difference in directional sensitivity resulting from transducer alignment and positioning of the hand-held probe. Furthermore, despite the efforts to avoid corrupted acquisitions during the data collection, experimental datasets still contain samples with relatively low signal-to-noise ratio. Such samples are expected to yield reduced performance metrics for image translation tasks due to significant mismatch between the predicted and noisy target images. In a clinical setting, a medical expert typically repeats an acquisition if they deem the signal quality is significantly lower than expected. However, beyond this subjective filtering step, one needs to make sure that even the worst results are either sufficiently good or their poor performance can be attributed to a cause. Accordingly, we further analyze some of the worst samples in the Appendix and believe that this should be a standard for future work. Provided dataset is limited to one body part of volunteers without any known health issues. The image reconstruction methods for OA imaging can be applied on any other body part or imaging setup as they solve the same physical inverse problem. The image translation algorithms for sparse acquisition and limited view problems can be adapted for different devices and acquisitions by another clinician/technician at another center, as the streak artifacts originating from sparse acquisition and limited view follow the same pattern.

We envision that future research will tackle additional challenges such as unsupervised or weakly supervised domain adaptation across the datasets provided in this work. There are initial studies to correct limited view artifacts in OA using transfer learning between simulated and experimental datasets after domain adaptation (Klimovskaia et al., 2022). Similarly, transfer learning between simulated and experimental domains can enhance segmentation performance of the vessels and skin curve in clinical images. Using properties of the detected tissues for fluence correction and heterogeneous SoS image reconstructions would then yield more accurate and quantitative images (Deán-Ben et al., 2014). We anticipate additional contributions in the field of representation learning using OADAT, e.g. through self-supervised learning, could allow overcoming bottlenecks for specialized downstream tasks with limited amount of task-specific available data. The multispectral datasets with paired images across multiple wavelength acquisitions are expected to facilitate the investigation of generative modeling of multispectral signals from a given wavelength. We also anticipate future work to explore novel multispectral unmixing approaches using MSFD, enabling more accurate quantification of oxygenation, melanin and lipid content of the tissues. Finally, the provided raw signal data, not available in commercial devices, has a high value for data-driven research. For example, it can serve to benchmark methodologies in image reconstruction e.g., based on variational networks with loop unrolling (Vishnevskiy et al., 2019), as well as ultra-fast imaging through adaptive channel sampling.

## 7    Conclusion

In this work, we provide experimental and synthetic clinical OA data covering a large variety of examples to be used as common ground to compare established and new data processing methods. The datasets correspond to samples from volunteers of varying skin types, different OA array geometries, and several illumination wavelengths. A subset of this experimental data is annotated by an expert. The dataset is supplemented with simulated samples containing ground truth acoustic pressure maps, annotations, and combine pairs of samples reconstructed with different OA array geometries. We define a set of 44 experiments tackling major challenges in the OA field and provide reconstructions of the images under these scenarios along with their corresponding ground truths. We propose and release 44 [5] trained neural networks that achieve a good performance for all these examples which can be used as baselines for future improvements. Additional problems can further be defined with the data provided, such as the effects of random sparse sampling or the presence of noise in the signal matrices prior to reconstruction. We believe that these datasets and benchmarks will play a significant role in fostering coordinated efforts to solve major challenges in OA imaging.

---

[5]Pretrained model weights and various scripts to train and evaluate modUNet are available at https://renkulab.io/gitlab/firat.ozdemir/oadat-evaluate.

**Broader Impact Statement**

The dataset is anonymized by randomly assigning identity numbers for each volunteer. The true identities will not be accessible by any third party now or in the future. The dataset should not be used to draw any medical conclusions. The purpose of the dataset is to help researchers to develop new image processing methods and provide benchmark scores.

**Author Contributions**

B.L. setup and acquired raw datasets, B.L. and F.O. analyzed and parsed datasets, F.O. conducted numerical experiments based on the defined tasks. All authors reviewed the manuscript.

**Acknowledgments**

This work was supported by Swiss Data Science Center (grant C19-04).

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
