# OpenReview forum: "OADAT: Experimental and Synthetic Clinical Optoacoustic Data for Standardized Image Processing"
_TMLR — Accepted by TMLR_

### Review · Reviewer_WrPA · 2022-12-14

**Summary Of Contributions:**

This paper aims to study the topic of  OA tomographic imaging. The authors provide both experimental and synthetic OA raw signals and reconstructed image domain datasets with different experimental parameters and tomographic acquisition geometries. And use trained neural networks to tackle three important challenges related to OA image processing.

**Audience:**

Yes

**Broader Impact Concerns:**

The Broader Impact Statement has been clearly stated, I have no extra concerns.

**Claims And Evidence:**

Yes

**Requested Changes:**

The authors need to make changes accordingly to address the concerns in the above section.

**Strengths And Weaknesses:**

Pros:
- The motivation and the importance of Optoacoustic (OA) imaging are well demonstrated.
- As an emerging method, the provided experimental data and simulations of forearm datasets are important and promising.
- The paper is well organized and the writing logic is clear.

Cons:
- The backprojection algorithm is used to generate OA images from the acquired signal. Are there any other algorithms that can be used? What are the differences?
- For sparse sampling, the sampled elements are fixed. Is it possible to make the sampling adaptive? i.e., using the different number of elements for different circle arrays.
- For the segmentation task, it would be better to also take the HD distance as one evaluation metric.
- For samples with a relatively low signal-to-noise ratio, besides repeating acquisition, are there any other technical solutions?
- On SWFD-mini, some results from the sparse view even outperform the full view. What could be the reason? More explanations are appreciated here.
- The UNet has already achieved good performance on different experiment tasks, the space for further improvements might be limited.

---

### Review · Reviewer_zakN · 2022-12-25

**Summary Of Contributions:**

The paper presented a set of optoacoustic datasets, both real and synthetic, with various image acquisition parameters and reconstruction methods that could be used for the development and evaluation of image analysis methods of optoacoustic imaging. The paper also provided three “model cards” to illustrate how the proposed data could be used in different tasks (i.e., reconstruction under limited view tomographic conditions, removal of spatial under-sampling artifacts, and anatomical segmentation for improved image reconstruction).

**Audience:**

Yes

**Broader Impact Concerns:**

The Broader Impact Statement in this work is sufficient.

**Claims And Evidence:**

Yes

**Requested Changes:**

1, Layout of table 2 should be revised

2, It is recommended that an extra table should be added to summarize the sample size, data constitution, and unique characteristics of the datasets presented in this work.


**Strengths And Weaknesses:**

Strength: large-scale, annotated dataset, as presented in this work, is always welcomed by the research community, especially as there exist few public datasets for optoacoustic imaging data.

Weakness: Table 2 is confusing to read; consider a more intuitive way of presenting the tasks.

---

### Review · Reviewer_LBfi · 2023-01-05

**Summary Of Contributions:**

The paper introduces a set of standardized datasets and tasks for machine learning processing of optoacoustic data. The paper introduces three tasks -- limited view correction, sparse sampling correction, and semantic segmentation – as well as four simulated and experimental datasets -- Multispectral forearm dataset (MSFD), Single wavelength forearm dataset (SWFD), Simulated cylinders dataset (SCD) and OADAT-mini a subset of 100 signal and corresponding reconstruction instances from each of the previously mentioned datasets. The paper concludes by reporting UNet-like model results for all three tasks ablating the type of input data, number of active transducer elements as well as the type of array used to acquire the signal.

**Audience:**

No

**Broader Impact Concerns:**

The Broader Impact is included in the manuscript. The reviewer has no comments here.

**Claims And Evidence:**

No

**Requested Changes:**

The paper is well written and provides a good overview of optoacoustic imaging. The details about data acquisition are interesting as well as the potential of the use of optoacoustic imaging in the clinic. Below the reviewer addresses the two acceptance criteria of TMLR journal.

**Are the claims made in the submission supported by accurate, convincing and clear evidence?**
In general, the paper does not make unsupported claims and the reviewer appreciates the standardization efforts in the OA field. However, there are some aspects of the submission that might require clarifications/corrections.
- From the text it is unclear to the reviewer why the three introduced tasks are truly the main challenges of OA. Further, the manuscript does not say why these three tasks are considered. Please clarify.
- To truly refer to the paper as a benchmark the reviewer would expect to see a comparison to simple methods in each task. In the introduction, the authors cover a variety of approaches for which the results could and should be reported in the manuscript. For example, sparse acquisition – “sparse or compressed sensing methods have been proposed both using conventional methods (Özbek et al., 2018) and deep learning algorithms (Davoudi et al., 2019).”--, limited view reconstruction – “new image processing pipelines have been suggested to improve limited-view-associated challenges in OA imaging by using data-driven algorithms in the image domain (Guan et al., 2020), signal domain (Klimovskaia et al., 2022) and combination of both domains (Davoudi et al., 2021; Lan et al., 2019).” – and Segmentation -- “segmentation of structures (Lafci et al., 2021; Schellenberg et al., 2022) in OA images has been shown to enhance the image reconstruction performance. Additionally, the optical fluence (intensity) also varies with depth across different tissues. This issue remains as one of the main factors affecting quantification in OA images (Brochu et al., 2017) and can also be corrected with tissue segmentation (Mandal et al., 2016).” --. Any reason why the introduced model is not compared to prior art?
- Any reason why the benchmark dataset is composed of relatively small datasets, e.g., “nine volunteers at six different wavelengths”, “14 volunteers at a single wavelength”? The datasets are large in the number of slices but small in the number of volumes.



**Would some individuals in TMLR's audience be interested in the findings of this paper?**
Although the applications of machine learning tools to the OA data could be of interest for the ML community and the introduction of datasets simplifies the application of ML tools to the OA data, the reviewer is not convinced whether the submission in question would generate interest in the TMLR community.
- First, please see the reviewer’s comments w.r.t. the choice of the tasks and the lack of comparisons with prior art. Please clarify this in your rebuttal.
- Second, the discussion of the reported results is rather short making it unclear to the reviewer what the added value to the community is, what the learnings of the performed data analysis are what the new generalizable learnings are. Please clarify this in your rebuttal.
- Finally, the OA data analysis might be too niche topic for the TMLR community and as such might not generate enough interest. From this point of view, the manuscript could be better suited for more OA oriented venue or medical imaging journal. To address this comment the authors could work on the presentation of the paper so that it becomes more accessible and as a result also more attractive to the broader ML/CV audience.


**Strengths And Weaknesses:**

Strengths:
- Well written manuscript.
- Good introduction to optoacoustic data acquisition.
- The discussed applications are motivated clinically.
- The idea of introducing standardization practices in the optoacoustic data model comparison is sound and well motivated.

Weaknesses (for details see below -- requested changes section)
- Unclear what are the key learnings of the performed analysis.
- The construction of the benchmark has missing motivation, and the benchmark is lacking comparisons to prior art.

---

### Decision · Action_Editors · 2023-02-16

**Recommendation:** Accept as is

**Comment:**

The main open comment in the final assessment is: "However, at the same time the authors fall short in providing a proper benchmarking of current and past approaches to the OA tasks -- they only provide one model and as they recognize in the rebuttal the results section provide little discussion, making it unclear what are the interesting observations for the TMLR community from this analysis...."

This is phrased as a matter of interest to the community, as the content of the experiments and benchmarking matches the claims of the paper.  As such, no additional changes are strictly required.  I would suggest to the authors that actively enabling benchmarking through the software release, e.g. by also hosting a leaderboard, might be a useful way of increasing potential impact.

**Audience:**

This is the main point of contention in the reviewing process.  It was the main reason for not recommending acceptance by Reviewer LBfi, and potentially limited interest was also highlighted by Reviewer zakN, who writes: "The concern from the reviewer is that optoacoustic imaging is a niche in the field of medical image analysis, which would limit its potential audience."

Quoting from the journal's documentation on acceptance criteria: " Would some individuals in TMLR's audience be interested in the findings of this paper?  This is arguably the most subjective criteria, and therefore needs to be treated carefully. Generally, a reviewer that is unsure as to whether a submission satisfies this criteria should assume that it does."

Although optoacoustic imaging is not very widely used at the moment, it is my assessment that research such as this paper is necessary to enable its adoption.  As such, my assessment is that the submission passes this criteria.

**Claims And Evidence:**

The problem addressed in the submission is in setting up a benchmark with a sound method for the evaluation of optoacoustic imaging (OI).  The main claim is stated in the introduction as follows: "Here, we provide experimental data and simulations of forearm datasets as well as benchmark networks aiming at facilitating the development of new image processing algorithms and benchmarking"

There is a statement: "Link to our datasets will be disclosed after acceptance to conserve anonymity." It is expected that the data, benchmark networks, and evaluation code will be released and of good quality.

---

> ### Author Response · Authors · 2023-02-24
> **Thank you**
>
> We thank the Action Editor for their decision to accept our manuscript as is.
> Following the decision, we have already updated the camera ready version to include all links and we released the dataset publicly. We envision https://github.com/berkanlafci/oadat as the entrance point to all relevant pointers for this manuscript. Nevertheless, we provide the following links for brevity:
> Data: https://www.research-collection.ethz.ch/handle/20.500.11850/551512
> Benchmark networks and evaluation code: https://renkulab.io/gitlab/firat.ozdemir/oadat-evaluate